# An evolutionary timeline of the oxytocin signaling pathway
Alina M. Sartorius[1,2], Jaroslav Rokicki[1,3], Siri Birkeland[4,5], Francesco Bettella [1,6], Claudia Barth [1,7], Ann-Marie G. de Lange[2,8,9], Marit Haram[10,11], Alexey Shadrin[1], Adriano Winterton[12], Nils Eiel Steen [1,7], Emanuel Schwarz [13,14], Dan J. Stein [15], Ole A. Andreassen [1,16], Dennis van der Meer[1,17], Lars T. Westlye [1,2,16], Constantina Theofanopoulou [18,19] ✉ & Daniel S. Quintana [1,2,16,20] ✉

Oxytocin is a neuropeptide associated with both psychological and somatic processes like parturition and social bonding. Although oxytocin homologs have been identified in many species, the evolutionary timeline of the entire oxytocin signaling gene pathway has yet to be described. Using protein sequence similarity searches, microsynteny, and phylostratigraphy, we assigned the genes supporting the oxytocin pathway to different phylostrata based on when we found they likely arose in evolution. We show that the majority (64%) of genes in the pathway are 'modern'. Most of the modern genes evolved around the emergence of vertebrates or jawed vertebrates (540 - 530 million years ago, 'mya'), including *OXTR*, *OXT* and *CD38*. Of those, 45% were under positive selection at some point during vertebrate evolution. We also found that 18% of the genes in the oxytocin pathway are 'ancient', meaning their emergence dates back to cellular organisms and opisthokonta (3500–1100 mya). The remaining genes (18%) that evolved after ancient and before modern genes were classified as 'medium-aged'. Functional analyses revealed that, in humans, medium-aged oxytocin pathway genes are highly expressed in contractile organs, while modern genes in the oxytocin pathway are primarily expressed in the brain and muscle tissue.

Oxytocin (OT) is a hormone and neuromodulator involved in a diverse plethora of functions in the central and peripheral nervous system across a wide range of species, such as parturition, metabolism, or social cognition (as detailed below). Homologs to *OXT* (the structural gene encoding OT) have been identified across many species from mammals to fish, suggesting that OT signaling is evolutionarily ancient and conserved. Although *OXT* orthologs have been traced for the first time in jawed vertebrate species' genomes[1], signaling pathways resembling OT/vasotocin signaling have already been reported in mollusks[2,3], insects[4,5], and tunicates[6]. Of note, *VT* (encoding the oligopeptide vasotocin) was identified to be the ancestral paralogous gene from where *OXT* was duplicated, based on their close proximity in the genomes of most vertebrate species, and on transposable elements that possibly drove the duplication of *OXT* from the *VT* locus ([1,7], "paralogous genes" refers to sibling genes belonging to the same gene family within one species, and they can emerge by gene duplication from a more evolutionarily ancient gene[7]). For instance, the involvement of OT in parturition has been demonstrated not only in a vast number of mammalian species (e.g., non-human primates[8], rabbits[9], and many others[10] (including vasotocin),[11,12]), but possibly also in non-vertebrates such

as annelids in the form of what has been considered distant homologs[13]. OT has also been implicated in the regulation of energy balance and metabolism in mammals[14–16]. There is thus a growing consensus that OT and structurally similar peptides are highly conserved, possibly reflecting conservation in their function[4,17–19].

In humans, OT has traditionally been associated with somatic reproductive processes, such as parturition[20] and lactation[21], but more recently it has further been linked to cardiovascular homeostasis, metabolism, and bone regeneration[22,23]. Its role in processes supporting social behavior and cognition, as well as decision-making and learning in general[24–26], has also become increasingly recognized in the past two decades[27]. As dysfunction in social behavior and cognition is a key feature of several psychiatric disorders and conditions, such as schizophrenia, it follows that OT's potential as a therapeutic to help alleviate social difficulties has been investigated. The results are, however, inconclusive thus far[28], which points to a persistent knowledge gap regarding the function and purpose of the human OT system and its treatment potential[29].

Genetic studies have aided in better understanding the role of the OT system in human biology and behavior. The most intensively studied genes

in the OT signaling pathway in both humans and non-human animals are the primary OT genes *OXTR*, encoding for the OT receptor, *OXT*, and *CD38*, which also regulates OT secretion[22]. However, more than 150 other genes are associated with the OT pathway as they enable and support OT signaling and functions, and mediate OT's effects on further agents and pathways like MAP kinases in humans, as reported in the established consensus encyclopedia on genes, genomes and pathways, "KEGG" ([30,31], detailed official pathway information available at https://www.genome.jp/entry/pathway+hsa04921). Among the sets of associated genes in the OT pathway is, for instance, the large family of genes coding for calcium voltage-gated channels, which facilitate fundamental processes in cell functioning and signal transduction, and accordingly support other pathways besides the OT signaling pathway, too. Research is yet to systematically determine when the ancestors of all genes supporting the OT signaling system, as opposed to two or three selected genes, emerged during evolution.

A characterization of the evolutionary history of all genes in the OT signaling pathway may guide the field toward a more comprehensive understanding of the system's current function and purpose in modern humans (see also ref. 26), since signaling pathways, like any other biological phenotype, are rooted in their evolutionary history and adaptiveness[32]. Evolution, ancestral environments and adaptions substantially contribute to shaping a phenotype into its current form and functionality, thus tracing back the history of the OT signaling pathway can help clarify the functions of the human OT system today. Moreover, pinpointing whether a signaling pathway consists primarily of ancient versus modern genes might be indicative of the specificity of a pathway. To this end, we sought to determine the time points in evolution when each of the genes in the OT signaling pathway emerged, and to identify evolutionary developments (e.g., the emergence of the central nervous system or milk provisioning) and gene expression patterns in humans that coincide with these time points. In this exploratory study, we applied a phylostratigraphic approach, using protein sequence similarity searches and microsynteny analyses—two methods commonly used in comparative and evolutionary genetics—to identify the evolutionary age of the primary OT genes and genes supporting OT signaling. We further tested whether genes in the OT pathway, having evolved around the emergence of a jawless and jawed vertebrate ancestor, showed signatures of positive selection during vertebrate evolution. Gene expression patterns were investigated by assessing the expression intensities of the genes in the OT pathway across human body tissues, including the brain, and by functionally annotating them. Providing an annotated evolutionary timeline for the OT signaling system substantiates and expands our current knowledge surrounding OT and helps better understand its function[26].

## Results

### The evolutionary age of genes in the OT signaling pathway
We used protein sequence similarity searches with protein BLAST ("BLASTp", https://blast.ncbi.nlm.nih.gov) and microsynteny analyses across different species to map the different homologs (orthologs or paralogs) of the OT pathway on the evolutionary timeline. Microsynteny analyses involved searching for the ten genes flanking our genes of interest to determine whether their genomic territory is evolutionarily conserved (see "Methods" for details). Protein sequence similarity and microsynteny searches are two of the most widely applied methods used in comparative genomics to identify homologous sequences across species[33,34]. When used in conjunction they can yield a robust estimate for potential orthology. BLASTp searches were used to identify homologs in 26 species before vertebrate evolution, and combined BLASTp and microsynteny searches were used to identify true orthologs in 13 vertebrates after vertebrate evolution. We refer to the former as homologs, since low genome and annotation quality in non-vertebrate species renders true orthology identification impractical. We used these approaches to gain insight into the evolutionary origin and age of the genes supporting the OT signaling pathway (Fig. 1).

Protein sequence similarity searches among genes facilitating OT signaling revealed that 18.2% of the genes had their earliest homologous

protein sequence in species dating back ~3500–1100 million years ([35]; see Supplementary Data 1, sheet 1, for an overview of the species included and sheet 2 for the BLASTp thresholds for invertebrates; see Supplementary Data 2 for BLASTp/microsynteny results for vertebrates). Thus, 28 out of the 154 genes of the OT pathway have homologs dating back to the first three phylostrata (PS), which we define as evolutionary "ancient". Twelve homologs (e.g., *EEF2*) were found in the first branch which includes prokaryotes (e.g., bacteria, archaea), 15 (e.g., *PPP3CC*) in the second branch of eukaryota (e.g., amoebozoa), and one (*PRKAG1*) in opisthokonta (third branch, e.g., fungi). For a different set of 28 genes (out of the 154 genes in total) we found meaningful protein sequence similarities in species representing PS 4 to 10, which we define here as "medium-aged". In other words, 18.2% of genes in the OT pathway appear to have homologous genes dating back about 1100–550 million years[35,36]. The seven PS include holozoa (e.g., choanoflagellata) with nine homologs (e.g., *CAMK2A*), metazoa (e.g., porifera/sponges) with two homologs (e.g., *ITPR1*), eumetazoa (e.g., cnidaria) with three homologs (e.g., *CCND1*), bilateria (e.g., mollusks, nematodes/roundworms, priapulida/priapulid worms) with eight homologs (e.g., *CAMK2*), deuterostomia (e.g., echinodermata, xenacoelomorpha) with three homologs (e.g., *ADCY5*), chordata (e.g., cephalochordata/lancelets) with one homolog (e.g., *KCNJ6*), and olfactores (e.g., tunicata) with two homologs (e.g., *RYR2*). In vertebrates, where we combined both protein sequence similarity and microsynteny searches, the majority of the genes in the pathway (63.6%, 98 out of 154 genes), including *OXT*, *OXTR* and *CD38*, had their earliest orthologs in different PS ranging from 11 to 20, which we define here as "modern" genes. For instance, we identified five orthologs in agnatha (that we take as a proxy for the jawless vertebrate ancestor, ~540 million years ago (mya,[36], PS 11), 34 (22.1%) orthologs in the great white shark (that we take as a proxy for the gnathostome/jawed vertebrate ancestor, ~530 mya[36], PS 12), 20 in amphibia (a proxy for the tetrapod ancestor, 440 mya[36], PS 14), two in xenarthra/afrotheria and laurasiatheria (a proxy for the eutherian and boreoeutherian ancestor, 180–140 mya[36,37], PS 17), and two in modern humans (a proxy for the homini clade, PS 20, see Supplementary Data 3 for all genes and the assigned PS). We found no new orthologs in PS 18 (euarchontoglires branch). These results reveal an accumulation in the emergence of genes facilitating OT signaling in the gnathostome ancestor (34 orthologs, 22.1%), 530 mya, whose most prominent extant representatives are sharks (Fig. 1). That is, approximately one fifth of the genes in the OT pathway seem to have originated in the jawed vertebrate ancestor.

Our comprehensive microsynteny analyses for the *OXT*, *OXTR*, and *CD38* genes revealed conserved syntenic territories across vertebrates after gnathostome evolution (~530 mya) for *OXT* (Fig. 2) and *CD38*, and after agnatha (jawless vertebrates) evolution for *OXTR* (~540 mya). This is in line with previous research identifying *OXT* and *OXTR* orthologs as vertebrate-specific[1,7,38–40], with *OXTR* appearing before the emergence of *OXT*.

### Signatures of positive selection in jawless vertebrate and gnathostome genes in the OT pathway
As described above, the combined BLASTp and microsynteny analyses revealed that approximately a quarter (39 out of 154) of the genes supporting the OT pathway emerged around jawless and jawed vertebrate evolution. That is, we found five orthologs in the sea lamprey, a proxy species for the vertebrate ancestor, and 34 orthologs in the great white shark, a proxy species for the gnathostome ancestor. In order to explore the evolutionary trajectory of these genes in more detail, we performed exploratory tests for positive selection across the same 13 vertebrate branches used for the BLASTp/microsynteny analyses leading up to the modern human in the 39 genes using adaptive branch-site random effects models ("aBSREL"[41], see also ref. 42). The tests produce, next to *P* values and test LRTs, a maximum $d_N/d_S$ value. The $d_N/d_S$ ratio quantifies the rate of synonymous to nonsynonymous substitutions and indicates selective pressures[43]. The resulting estimate can range from 0 to, theoretically, infinity.

Of the 38 genes tested (originally 39, one gene excluded during analysis, see "Methods" for details), 17 (44.74%) were found to be under positive

selection in at least one branch or node (i.e., a branching point where a branch splits into two new lineages from a last common ancestor[44], see Supplementary Data 4 for all results across branches and nodes). The node or branch with most accumulation of genes under positive selection was the "mammalian" Node 8 branching into the two lineages leading to prototheria (represented by the platypus) and theria with five significant genes (*ADCY2*: $P = 1.94e-2$, *CACNA2D1*: $P = 2.06e-3$, *EGFR*: $P = 1.30e-5$, *MEF2C*: $P = 1.64e-6$, *RYR3*: $P = 2.37e-2$). In addition, in the order of evolutionarily most ancient to modern vertebrate branches and nodes, two genes were found to be under positive selection in the branch leading up to the great white shark

(*NFATC1*: $P = 6.74e-3$, *RYR3*: $P = 2.37e-2$), one gene in the branch leading to the zebrafish (*RYR3*: $P = 1.24e-3$), two genes in the "tetrapode" Node 5 that splits into amphibia (as represented by the western clawed frog) and amniota (*EGFR*: $P = 4.77e-2$, *PPP1R12A*: $P = 1.81e-2$), and two genes in the chicken/red junglefowl ancestral branch (*ADCY7*: $P = 2.85e-3$, *NFATC1*: $P = 8.09e-4$). The adaptive branch-site test further revealed that, in the branch leading to the platypus, one gene showed significant signatures of positive selection (*ADCY8*: $P = 5.01e-4$), three genes showed signs of positive selection in the "therian" Node 9 branching into marsupialia (represented by the Tasmanian devil) and eutheria (*ADCY8*: $P = 3.67e-2$, *EGFR*:

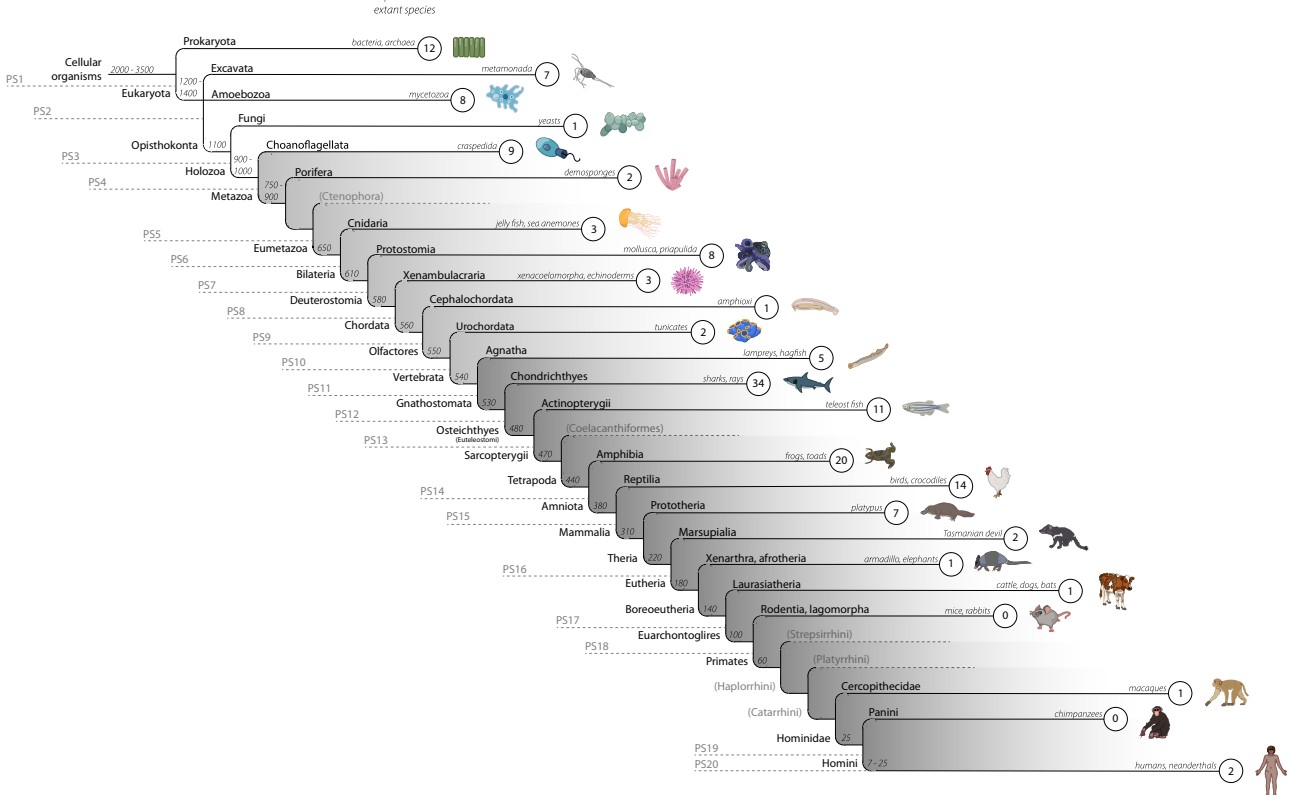

**Fig. 1 | The evolutionary age of the OT signaling pathway.** The emergence of genes supporting the OT signaling pathway is outlined on a simplified phylogenetic tree tailored to human evolution, thus starting with cellular organisms and ending with modern humans. The clade name and an example of a representative extant species of the respective lineage ancestor are shown for each branch. The absolute gene counts per phylostratum (PS, 1 = oldest, 20 = newest) are given in the smaller circles at the ends of each branch. Branches with ancient genes are highlighted in white, branches with medium-aged genes in light gray and modern genes in gray. Time estimates for lineage splits (e.g., 650, 310, 60) are given in mya ("million years ago"). For example, three genes supporting the OT signaling pathway had their earliest homolog emerge in the PS 8 ancestor (~580 mya), for which the extant echinoderm purple sea urchin (*S. purpuratus*) is the proxy species, which suggests these three genes in the pathway emerged in the deuterostomia ancestor.

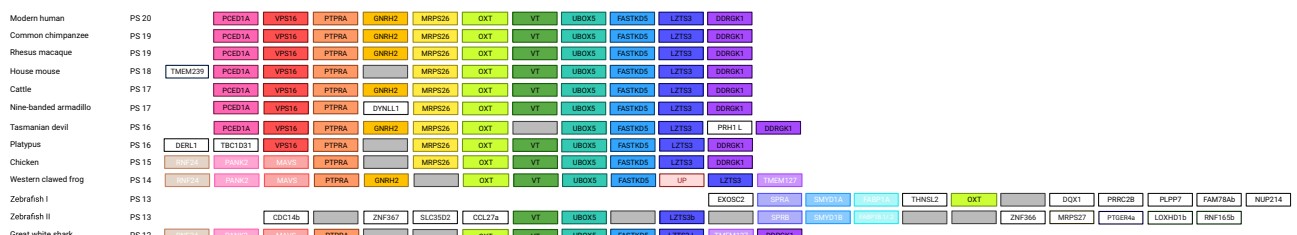

**Fig. 2 | Microsynteny for *OXT* across PS 20–12.** The ten genes surrounding *OXT* in the modern human build a microsyntenic block that is conserved across mammalian species. The synteny is less conserved in teleost fishes, as shown in the zebrafish microsyntenies for *OXT* and *VT* in two different loci (Zebrafish I and II), possibly due to a whole-genome duplication in the teleost fish ancestor[134] that gave rise to genome reorganization events. The species' common names are given in the outer left column, followed by the corresponding PS and the microsyntenic block. Each rectangle represents one gene with the abbreviated gene name in the center. Gray, empty rectangles indicate a potential missing gene in that locus. Non-colored rectangles indicate no re-occurrence of a gene in the displayed species within microsyntenic distance. The microsyntenies for *CD38* and *OXTR* are visualized in Supplementary Figs. 1 and 2. PS phylostratum, UP uncharacterized protein.

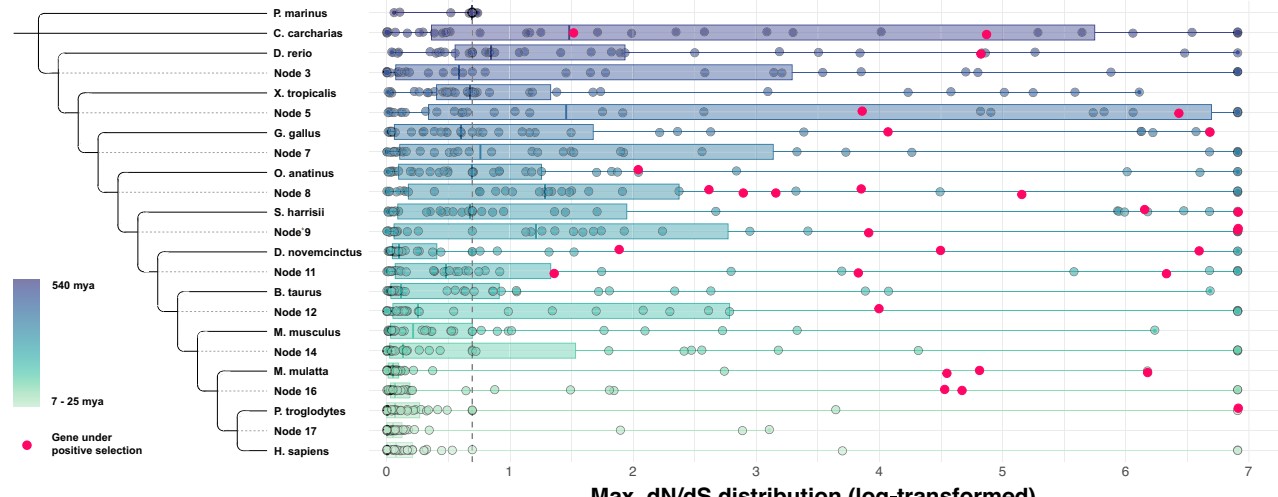

**Fig. 3 | Distribution of maximum $d_N/d_S$ values from the adaptive branch-site random effects likelihood (aBSREL) models.** The maximum $d_N/d_S$ ratios resulting from the aBSREL models for $n = 38$ (39) genes across 13 vertebrate species and ten nodes. Each dot represents one gene, the dots highlighted in magenta are the genes that were under positive selection in specific branches or nodes as identified in the branch-site tests. Branches and nodes are displayed on the $y$ axis and the maximum log-transformed $d_N/d_S$ values on the $x$ axis. log-$d_N/d_S$ ratio < 0.693147181 signifies negative selection, log-$d_N/d_S$ ratio = 0.693147181 signifies neutral selection, log-$d_N/d_S$ ratio > 0.693147181 signifies positive selection. The vertical dotted gray line marks neutral selection. The center line in each box plot indicates the median, box limits are the upper and lower quartiles, whiskers reach up to the minimum and maximum value, respectively. The raw data underlying this plot is presented in Supplementary Data 5.

$P = 4.77\text{e-}2$, $RYR3$: $P = 8.32\text{e-}3$), two genes in the Tasmanian devil ancestral branch ($ELK1$: $P = 4.64\text{e-}3$, $OXTR$: $P = 2.81\text{e-}2$), three genes in the "eutherian" Node 11 that splits into xenarthra/afrotheria (represented by the nine-banded armadillo) and boreoeutheria ($ADCY7$: $P = 6.82\text{e-}3$, $CACNA2D3$: $P = 4.03\text{e-}2$, $EGFR$: $P = 4.07\text{e-}2$), and again three genes in the branch leading to the nine-banded armadillo ($CACNA2D4$: $P = 3.73\text{e-}4$, $CD38$: $P = 8.66\text{e-}3$, $NFATC3$: $P = 1.28\text{e-}15$).

Moreover, a proportion of nucleotide sites of one gene was found to be under selection in the "boreoeutherian" Node 12, which branches into laurasiatheria (represented by modern cattle) and euarchontoglires ($CACNA2D1$: $P = 2.83\text{e-}2$), of two genes in the "primate/catarrhini" Node 16 (splitting into cercopithecidae, including the rhesus macaque, and hominidae; $ELK1$: $P = 2.06\text{e-}2$, $PLA2G4A$: $P = 3.03\text{e-}7$), and of three genes in the rhesus macaque ancestral branch ($MEF2C$: $P = 2.00\text{e-}10$, $PLA2G4A$: $P = 4.31\text{e-}3$, $PRKAB1$: $P = 1.12\text{e-}9$). Lastly, one gene, $PLA2G4A$, was found to be under strong positive selection in the branch leading to the common chimpanzee ($P < 0.0001$; all above-mentioned $P$ values were FDR-corrected for multiple comparisons). The maximum $d_N/d_S$ values for each gene in each branch and node are visualized in Fig. 3. The more modern the tested branch or node becomes, the narrower the distribution of maximum $d_N/d_S$ values appears, suggesting that extreme values are less common in more modern branches and nodes.

**The functional relevance of age estimates for genes in the OT pathway**
Our evolutionary analysis of the genes in the OT signaling pathway revealed 28 ancient genes (PS 1–3), 28 medium-aged genes (PS 4–10), and 98 modern vertebrate genes (PS 11–20, Supplementary Data 3). To investigate the functional relevance of these gene subsets in the OT pathway, these three gene sets were submitted separately to FUMA ("Functional Mapping and Annotation of Genome-Wide Associations Studies"[45]), which integrates human gene expression data from the GTEx database and GWAS data. This analysis revealed that out of 30 tissue types across the body, genes from the medium-aged gene set were significantly upregulated in blood vessels (default Bonferroni corrected $P = 3.61\text{e-}7$) and the bladder (default Bonferroni corrected $P = 1.26\text{e-}3$), and genes from the modern gene set were significantly upregulated in skeletal muscle tissue (default Bonferroni corrected $P = 1.46\text{e-}7$) and the brain (default Bonferroni corrected $P = 1.89\text{e-}3$,

Fig. 4). The modern gene set in the OT pathway was further enriched in two GWAS associated with tooth decay and denture use (both $P = 3.72\text{e-}2$, FDR-corrected for multiple testing), which corresponds to the evolution of the jaw, and mineralization of teeth and bone in vertebrate evolution (see below). In another GWAS they were found to be related to "Automobile speeding propensity" (also $P = 3.72\text{e-}2$, FDR-corrected), which could be considered a proxy for impulsivity[46]. The ancient genes supporting the OT pathway showed no tissue-specific upregulation or enrichment in any GWAS, with the latter being the case also for medium-aged genes with any GWAS (see Supplementary Data 6–8).

**Cerebral expression of modern genes supporting OT signaling**
Based on our findings that modern genes in the OT pathway were differentially expressed in the brain, we next explored gene expression differences between brain regions in this subset of genes from the OT pathway. Between-brain region comparisons of the expression patterns of the gene set were calculated for 42 brain regions with data from the Allen Human Brain Atlas ("AHBA" dataset, https://human.brain-map.org, parcellation based on the Desikan–Killiany atlas, focus on left hemisphere due to a larger sample size, cortical and additional subcortical regions included). The region-specific mRNA levels of the modern gene set in each of the 42 brain regions were compared to the average, whole-brain mRNA intensity of the modern gene set in the pathway. This yielded statistically significant mean differences in four cortical brain regions (pars opercularis: $P = 4.11\text{e-}2$, $d = 1.75$, $t = 4.29$; posterior cingulate: $P = 4.11\text{e-}2$, $d = 1.79$, $t = 4.38$; precentral gyrus: $P = 1.06\text{e-}2$, $d = 2.98$, $t = 7.29$; superior frontal gyrus: $P = 4.44\text{e-}2$, $d = 1.67$, $t = 4.08$; all $P$ values FDR-corrected), and five subcortical regions (thalamus proper: $P = 5.24\text{e-}4$, $d = 6.98$, $t = -17.11$; pallidum: $P = 2.27\text{e-}2$, $d = 2.36$, $t = -5.79$; accumbens area: $p = 4.11\text{e-}2$, $d = 1.81$, $t = 6.87$; hippocampus: $P = 4.11\text{e-}2$, $d = 1.77$, $t = -4.35$; brainstem: $P = 1.74\text{e-}3$, $d = 4.74$, $t = -11.61$; all $P$ values FDR-corrected). This indicates that the modern genes supporting the OT system are upregulated in certain cortical regions, most pronounced in the precentral gyrus, and down-regulated in certain subcortical, central structures, such as the basal ganglia and limbic structures, as well as in the brainstem (Fig. 5, for all test statistics see Supplementary Data 9). Given the heterogeneous sample characteristics of the AHBA dataset, we assessed the consistency of the cerebral expression patterns we identified for the genes in the OT pathway across donors using

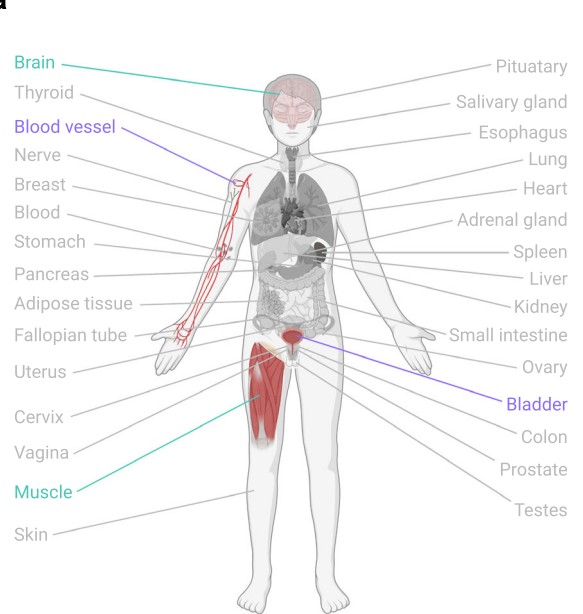

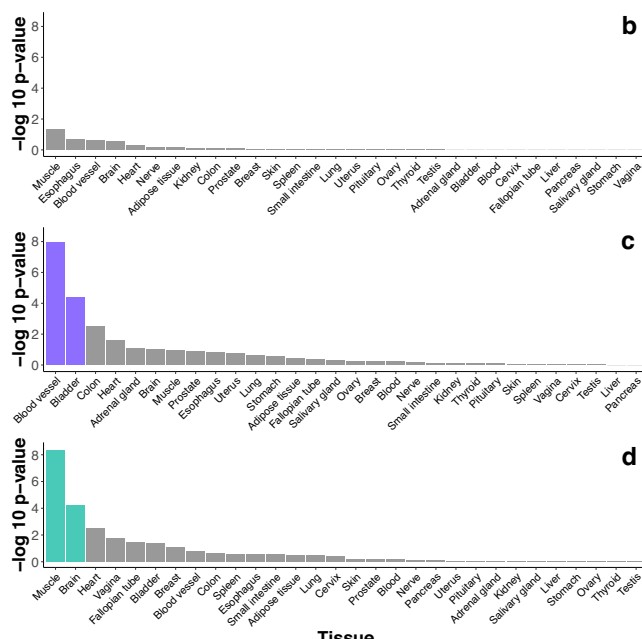

**Fig. 4 | Expression of the genes in the OT pathway across the body. a** Differential expression of genes from the OT signaling pathway in 30 tissue types from the GTEx dataset. Significantly enriched tissues are colored, with colors corresponding to the bar plots to the right. **b** Ancient OT pathway genes ($n = 28$) were not significantly differentially expressed (e.g., up- or downregulated) in any tissue. **c** Expression of the medium-aged genes ($n = 28$) is enriched in blood vessels and the bladder. **d** Modern genes ($n = 98$) are upregulated in skeletal muscle tissue and the brain. —Log 10 $P$ values from hypergeometric tests. The raw data underlying plots (**b–d**) is presented in Supplementary Data 11.

the differential stability metric, as implemented in the abagen toolbox[47]. Differential stability (DS) is a measure based on pair-wise averaged Pearson correlations which can be used to quantify expression reproducibility[48]. We found that 77.21% of the genes in the OT pathway (105 out of 136, no data available for 18 genes) were among the top 50% of all protein-coding genes in terms of expression stability, which included *OXT*, *OXTR* and *CD38*. This is indicative of relatively stable expression intensities of these genes in the OT pathway across donors within a comparable age category (Supplementary Data 10).

## Discussion

Comparative and evolutionary genetics can shed light on the biological relevance of genes and the phenotypes they support. With our open, exploratory approach and analyses, we identified the evolutionary time points when genes associated with the OT signaling pathway arose, with most of them having emerged in the gnathostome (jawed vertebrate) ancestor. Functional annotation of these genes demonstrated that medium-aged genes are highly expressed in human contractile organs (e.g., blood vessels and bladder), whereas modern genes are upregulated in muscle tissue and in the brain, especially in cortical regions.

Combined protein sequence similarity searches and microsynteny analyses revealed that almost 20% of the genes in the OT pathway are ancient, as they were found in species representing the first three PS (e.g., bacteria, yeasts, and other fungi phyla[49]). Some of these genes play a critical role in basal cellular mechanisms and organism functioning, as they encode, for instance, the RAS protein family (e.g., *KRAS*, *NRAS*) or are members of the protein serine/threonine phosphatase family (e.g., *PPP3CC*, *PPP3R1*). Others encode serine/threonine-specific protein kinase enzymes (e.g., *PRKAA1*, *CAMK1*), which are involved in many integral intracellular signaling processes[50–52]. Among the medium-aged genes of the OT pathway that emerged in PS 4–10 (holozoa to olfactores), we identified a gene family coding for the voltage-gated L-type calcium channel (e.g., *CACNB2*, *CACNB4*), an ion channel that is crucial for numerous cellular functions and may play a role in cardiac arrhythmia[53]. Most of the genes in this age

category accumulate in PS 4 (~32%)—and thus likely evolved in the holozoan ancestor—and in PS 7 (~29%)—and could thus have evolved in the bilaterian ancestor. Choanoflagellates (e.g., *S. rosetta*) are prominent representatives of PS 4, which are thought to be the bridge connecting unicellularity with multicellularity[54]. PS 7, including species like mollusks and priapulida/priapulid worms, is where it is thought that a centralized nervous system—which was likely not present in the eumetazoan pre-decessor—first appeared[55]. We found that among the modern genes facilitating OT signaling, various genes from the voltage-dependent calcium channel γ family were highly expressed in cerebral tissue[56,57]. Interestingly, subunits *CACNG*2-4, 7, and 8, along with *OXT* and *OXTR* and other genes in the OT pathway, account for the brain expression signal we found in our functional annotation FUMA analyses. *CACNG4*, *5*, *6*, and *8*, which are gene members of the same family, have been associated with schizophrenia risk[58]. Within that subset of modern orthologs in the OT pathway, more than one-third were allocated to the evolution of the jawed vertebrate ancestor (represented by the great white shark) in PS 12 (Fig. 1). The emergence of the jawed vertebrates (gnathostomes) was a major evolutionary milestone and is characterized by several gene duplications, most likely including the origin of *OXT* from *VT*[1,7,59,60]. On a phenotype level, the development of a bony jaw, sophisticated teeth, mineralization of different bone structures, the development of an adaptive immune system, and the emergence of the para-sympathetic nervous system, among other features, were important advancements that likely facilitated the diversification of feeding and preying behavior[61,62]. The OT system plays an important role in processes relevant to these phenotypes, such as energy balance regulation and metabolism[22,63], osteoblast regulation[64,65] and bone maturation and preservation[22,66]. Whether these OT functions markedly shaped gnathos-tome evolution or whether the observation is coincidental needs to be further substantiated by more research, as microsynteny analyses alone cannot determine this. Although speculative, broader dietary options could have led to further development of the central nervous system and an increase in brain size, as a similar connection has already been hypothesized for the human lineage[67]. Lastly, *OXT* and *CD38* had their earliest ortholog in

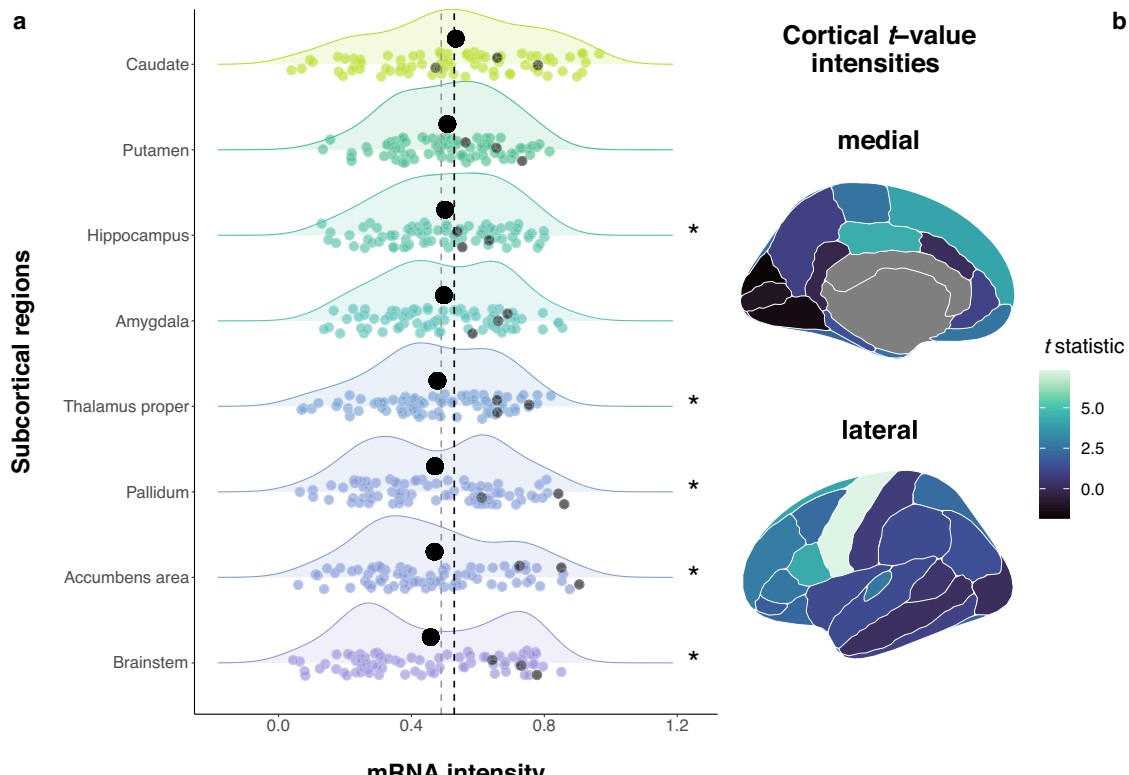

**Fig. 5 | Expression of the modern OT pathway genes across the human brain.**
**a** Mean expression values of the modern OT gene set in eight subcortical regions
(bold black dots) compared to the average expression of the modern OT gene set
across the whole brain (i.e., the total of 42 brain regions, cortical and subcortical,
vertical black dashed line; for reference the vertical gray dashed line represents mean
expression across subcortical regions). Jittered colored dots represent the mRNA
intensity of each single modern OT gene, the three gray dots in each distribution

represent mRNA intensities of *OXT*, *OXTR* and *CD38*. *$P_{FDR}$ < 0.05. Ridge plots are
colored and ranked by mean expression intensity. **b** Atlas representation of the *t*
values in 34 cortical brain regions (unilateral left). High *t*-statistics can be observed in
the precentral gyrus, posterior cingulate, pars opercularis, and superior frontal gyrus
(statistically significant), the lowest value appears in the cuneus. The raw data
underlying these plots is presented in Supplementary Data 12 and 13. *n* = 81 (98)
genes and postmortem expression data from *n* = 6 brains included in the analysis.

PS 12 (gnathostomata), whereas we found an ortholog for *OXTR* in PS 11
(vertebrata), before *OXT*, in line with findings from previous research on the
origin of *OXT* and its receptor[1,7,40,59,68].

   To summarize, many of the ancient and medium-aged genes, but also
some of the modern genes that facilitate OT signaling are not exclusively
part of the OT pathway. They support a variety of different pathways due to
their function in ubiquitous cellular mechanisms. This, in conjunction with
the finding that some genes in the OT pathway emerged several million
years before the primary OT genes themselves, points to a scenario where
already existing proteins and genes became involved with *OXT* and *OXTR*
signaling and regulation at a later point in time during vertebrate evolution.
They then progressively began to work in concert with and facilitate the
more recently evolved OT nonapeptide system around *OXT* and *OXTR*
signaling. Subsequently, other more modern genes that evolved after *OXT*
and *OXTR* may have taken part later in the polygenic pathway that had
already been established in early gnathostome evolution.

   Based on the observation that just over 25% of the genes facilitating the
OT pathway seem to have had their earliest ortholog around the time of the
emergence of the jawless and jawed vertebrates, we assessed whether
selective pressures affected those genes particularly during the remaining
vertebrate evolution up until the emergence of homini. Our exploratory
positive selection analyses revealed evidence for positive selection in 17 out
of 38 tested agnathan (jawless vertebrate) and gnathostomian (jawed ver-
tebrate) genes in one or more of the 13 branches and/or ten nodes during the
last 540 million years. Of the principal OT genes, *OXTR* was under positive
selection in the marsupialia branch leading to the Tasmanian devil, and
*CD38* in the xenarthra/afrotheria branch leading to the nine-banded
armadillo. The signal of positive selection of *OXT* in the homini branch

leading to the modern human did not survive correction for multiple testing.
Signatures of positive selection of multiple genes were especially prominent
in the 'mammalian' node which branches into the prototheria lineage
(represented by the extant monotreme platypus) and the theria lineage
(leading to among others the Tasmanian devil, cattle, rodents, and pri-
mates), 310 million years ago. Interestingly, a sophisticated lactation system
has first been documented in early mammals like prototheria and theria[69].
The node further leads to the lineage split between the last egg-laying
mammals (e.g., prototherians) and viviparous mammals (i.e., eutherians
and marsupials). The mammalian node hence possibly explains the genetic
architecture of an important milestone in the evolution of lactation and milk
provision for offspring, which is a key differential feature of mammals. As
mentioned, *OXTR* was under positive selection in the marsupialia branch.
Marsupials are reported to have a more complex and highly adaptive lac-
tation system since the milk composition can change drastically in some
species throughout the lactation cycle[69,70]. Hence, the positive selection of a
considerable portion of the modern genes supporting the OT system during
the evolution of the mammalian lactation system is consistent with OT's
established role in milk let-down[21]. The two genes that were under positive
selection in most branches or nodes were *EGFR* and *RYR3*. *EGFR* encodes
the epidermal growth factor receptor and has been largely implicated in the
development of different types of cancers[71,72], and has been explored as a
treatment for neuropathic pain in cancer patients[73]. It may also be relevant
for non-pathological physiological functions (e.g., tyrosine kinase activa-
tion, tyrosine autophosphorylation[74]). *EGFR* seems to be involved in OT
signaling in several ways. One of them is the mediation of the anti-
proliferative effect of OT[75], which could connect back to *EGFR*'s relevance in
cancer. Furthermore, *EGFR* could be implicated in the OT pathway via

regulating OT-triggered prostaglandin pulses in the endometrium ([76], as shown in bovines). The functions of the gene *RYR3* (coding for the ryanodine receptor 3) are less straightforward since it has been linked to a variety of conditions, ranging from hypertension to diabetes[77,78].

When genes are found to be under positive selection (i.e., a proportion of the nucleotides building the gene shows signatures of positive selection), this can be indicative of the genes being 'favored' (i.e., selected) during a particular period of time in evolution. It follows that the 17 genes here were of likely importance and underwent evolutionary adaption during specific periods in time, possibly because they supported phenotypes that were beneficial for survival and reproduction during that time.

Our gene enrichment and functional annotation analyses pointing to medium-aged genes in the OT pathway being primarily enriched in blood vessel and bladder tissue could correspond to their function in tissue constriction and contraction. The enriched expression of the modern genes in the OT pathway in the brain and muscle tissue (Fig. 4), compared to the other tissue samples, could correspond to the role of OT in uterine, heart, and mammary cell contraction[79–81], as well as to an association with phenotypes related to cardiovascular and energy regulation[79], such as body weight[14,82], and cognition[24,25]. These results substantiate the role of the primary OT genes and other genes involved in OT signaling in shared somatic functions across vertebrates and invertebrates, as well as in higher-order functions of the central nervous system in vertebrates, which presumably evolved later.

Our detailed cerebral gene expression results shed light on the specifics of the upregulation of modern genes in the OT pathway in the brain, showing that those genes are significantly more expressed in certain cortical regions, mostly frontal lobe regions (e.g., superior frontal gyrus, precentral gyrus, Fig. 5). Since cortical, particularly fronto-cortical, regions are involved in higher-order cognitive functions[83,84] and partially motor control[85], this could link to the role of the OT system in cognition (e.g., ref. 24) and neurodevelopmental conditions associated with motor control challenges[86]. The downregulation of the modern genes in the signaling pathway in subcortical regions might at first glance appear contradictory to other studies. For example, Quintana and colleagues[87] reported an increased *OXTR* and *CD38* expression in the thalamus. However, it is important to note that in our study, out of the modern genes tested, *OXTR* and *CD38* were among the above-average upregulated genes (whole-brain (black dashed line) and subcortical (gray dashed line) average) in all of the subcortical structures, especially in the pallidum, accumbens area, caudate, and thalamus proper (Fig. 5a). We also found some *OXT* mRNA signal in subcortical regions that are usually not prominently associated with the synthesis of *OXT*. The presence of small amounts of *OXT* mRNA in those regions is also reported in other databases (e.g., "Bgee"[88], https://www.bgee.org; "GTEx"[89], https://gtexportal.org) and studies (e.g.,[87], using the same data), and the *OXT* mRNA intensity is still considerably higher in regions where it is to be expected (e.g., hypothalamus, compare https://human.brain-map.org). Lastly, the results of the cerebral gene expression analysis rely on the high-resolution AHBA brain expression dataset. Although this dataset provides the major advantage of higher anatomical resolution, it only consists of six donors, with mixed ancestries and an unbalanced sex ratio. To address this heterogeneity in the sample, we used differential stability (DS) which showed that almost 80% of the signaling pathway genes had DS scores that were among the top 50% of all protein-coding genes, indicating generalizable gene expression patterns[48].

There are additional limitations to the current study that are worth noting. First, an issue that hampers the identification of true orthologs in more ancient species is the lack of fully sequenced, annotated, complete and high-quality genomes for many invertebrate species. Genomes of thoroughly studied ancient model organisms like *C. elegans*, *E. coli* and *S. cerevisiae* are available. However, genomes and proteomes of ancient species stemming from PS 4–6, for instance, are often only sequenced at a scaffold level and are still incompletely annotated. Data availability and quality improve slightly for species of PS 7–11. These circumstances render the search for conserved syntenic blocks across species intricate.

Consequently, the emergence of orthologous genes of the current OT signaling pathway in older species cannot be reliably traced back, hence we attributed as "homologs" the genes we identified based on BLASTp hits only. One should bear in mind that we cannot rule out that these homologs could either be orthologs (i.e., the same gene across species) or paralogs (i.e., genes of the same gene family but not the same gene). The search for orthologous genes will remain a challenge until genomes from species older than vertebrates are sequenced at a chromosome level and properly annotated with a uniform consensus gene nomenclature[90]. Furthermore, the emergence of vertebrates is characterized by one or several rounds of whole-genome duplications (which also produced the diverse *OXT/VT* gene family in vertebrates), which might be linked to the origin of evolutionary novelties[91,92]. Since plenty of new genetic material seems to have emerged around that time, it remains to be clarified whether the "gnathostomian" boost of novel genes supporting the OT pathway nowadays is part of this general genomic proliferation, or whether it is a unique occurrence specific to the OT system. Future research on other endocrine signaling pathways of similar size using the same analysis pipeline may help resolve this question. Regarding the positive selection analysis, we deployed aBSREL as implemented in HyPhy[41] to test for positive selection in a non-hypothesis-driven fashion to accommodate the exploratory nature of the analysis, and leveraged the advantage of the method to automate running a large number of tests for multiple branches and nodes simultaneously without the requirement to specify foreground branches a priori. However, it must be acknowledged that this analysis approach necessitating multiple testing of several branches has reduced statistical power and thus an increased chance of false negatives. Concerning the FUMA analysis, we submitted three gene sets of varying sizes ($n = 28$, $n = 28$, $n = 98$). Since the online tool is based on hypergeometric tests operating in the background, it must be noted that smaller gene sets, like the ancient and medium-aged gene sets in this study, are more likely to produce false positives, as compared to larger gene sets, such as the modern gene set in this study. Looking at the accumulated published studies using FUMA, there currently does not seem to be a standard corrective mechanism for this potential issue in place. The investigation of such a mechanism may be of interest to future studies. Lastly, for the cerebral gene expression we used data from the left hemisphere only due to the reduced sample size for the right hemisphere in the AHBA dataset. The human brain is known for its lateralization, and asymmetries in transcriptomes[93]. Thus, the reported gene expression patterns might be slightly different for the right hemisphere. Until more high-resolution parcellation data for cerebral gene expression is available from a larger sample size in both hemispheres, this issue remains to be clarified.

In this study, we found that homologs enabling OT signaling emerged in different PS throughout evolution, with the vast majority of them originating after the emergence of the vertebrate ancestor. We confirm prior studies reporting that OT ligand orthologs date back to at least 530 mya[4,19,59], and our analysis expands those findings, suggesting that ~40% of homologs of genes supporting what makes up the OT signaling pathway date even further back[49,94]. Accordingly, we suggest that the evolution of the OT signaling pathway was a gradual process in which evolutionary ancient genes first started interacting with and supporting OT signaling around the evolution of vertebrates and jawed vertebrates, but other genes joined the OT signaling system afterward. We also provide evidence that modern genes part of the OT signaling pathway, found only in vertebrates, are significantly expressed in human muscle and brain tissue and have been associated with mental illness and complex cognition. The role of the OT system in non-stereotypical locomotion, higher-order cognition and social behavior may therefore not have been an initial function but only have evolved later when new environmental demands required it. We also provide a possible link between the emergence of phenotypic novelties related to the jaw and bones seen in gnathostomes and the lactation system evolved in early mammals, and the OT pathway. By systematically mapping the evolutionary history of the OT signaling system, and gene expression correlates in humans, our study provides a greater understanding of the OT signaling pathway that can inform future experiments and treatments in humans.

## Methods

If not stated otherwise, the statistical software R (version 4.2.0[95]) and RStudio (version 2022.7.1.554[96]) were used for analyses and data visualizations (except for Figs. 1, 2, parts of Figs. 3, 4a, see "Acknowledgments"). The R package `tidyverse`[97] was used to conduct core analyses (see below and supplementary references for further R packages used). The 154 genes thought to be involved in OT signaling were retrieved from the official consensus encyclopedia "Kyoto Encyclopedia of Genes and Genomes" (KEGG[30]), where they are reported to be part of the OT signaling pathway. An overview of the pathway information and visual representation is accessible at https://www.genome.jp/entry/pathway+hsa04921, see also ref. 31.

### Gene age estimates (phylostratigraphy)

To identify gene orthologs (of the 154 genes currently supporting the OT signaling pathway) in *vertebrates*, we performed protein BLAST ("BLASTp", NCBI, https://blast.ncbi.nlm.nih.gov) queries and microsynteny analyses. The *Homo sapiens* protein sequence (FASTA format, downloaded from https://blast.ncbi.nlm.nih.gov) was used as a reference and queried against the proteomes of the following target species: chicken/red junglefowl (*Gallus gallus*), western clawed frog (*Xenopus tropicalis*), zebrafish (*Danio rerio*), great white shark (*Carcharodon carcharias*), and sea lamprey (*Petromyzon marinus*; database "non-redundant protein sequences"). These five species, as extant descendants of major vertebrate lineages, were taken as proxies for their amniota, tetrapoda, euteleostomi, gnathostomata, and agnatha/jawless vertebrates ancestor, respectively. They are furthermore well-studied model organisms and have genomes sequenced at chromosome level (i.e., high coverage) available. From the BLASTp results, we used the hits (and their accession numbers) with the lowest E-value (expect value) and highest max score of each target species in each gene and entered them into the NCBI gene search engine (https://www.ncbi.nlm.nih.gov) to trace back the gene underlying the hit. Subsequently, we used these gene loci to perform microsynteny analyses on a 10-gene window, that is, the five protein-coding genes before and after our gene of interest. Neighboring protein-coding genes were identified using the information from the "genomic regions, transcripts, and products" section in the NCBI gene database; pseudo-genes, exons, and RNA regions were excluded. Vertebrate microsynteny for each gene supporting the OT signaling pathway was established using this approach. For those genes where the synteny of neighboring genes around the focal gene seemed to be markedly scattered or absent in the sea lamprey or great white shark, we hypothesized that these genes had evolved after agnathans/jawless vertebrates or gnathostomes and no further BLASTp searches with *non*-vertebrate species were run. For those "vertebrates genes" that were present in the modern human genome but appeared to have no shared synteny in the chicken/red junglefowl, western clawed frog or zebrafish either we searched for BLASTp hits and microsynteny in other mammalian species to determine the timing of their emergence (i.e., chimpanzee (*Pan troglodytes*), rhesus macaque (*Macaca mulatta*), house mouse (*Mus musculus*), cattle (*Bos taurus*), nine-banded armadillo (*Dasypus novemcinctus*), Tasmanian devil (*Sarcophilus harrisii*), and platypus (*Ornithorhynchus anatinus*); see Supplementary Data 1, sheet 1, for a list of the vertebrate species).

We then performed BLASTp searches in 26 *non*-vertebrate target species for the genes with vertebrate-wide microsynteny that was present in the sea lamprey or the great white shark ($n_{gene}$ = 95). We used sea lamprey or great white shark protein sequences identified before of these remaining genes as queries and BLASTed them against the proteomes of the 26 target species ranging from cellular organisms (e.g., bacterium *Escherichia coli*) to tunicates (e.g., *Ciona savignyi*; see Supplementary Data 1, sheet 1, for a list of the invertebrate species). This was done to help determine whether there are any potential homologs in more ancient species than vertebrates. Of note, since microsynteny is less feasible and reliable for the majority of ancient species' genomes, due to poor sequencing quality and incomplete annotation, no additional microsynteny analyses were performed in the non-vertebrate species. Possible hits in non-vertebrates are therefore referred to as "homologs" and could not be distinguished between "orthologs" and

"paralogs". A scaled *and* averaged max score threshold specific to each of the 26 non-vertebrate species was calculated as follows to function as a cutoff value to determine reliable protein sequence similarity. For each protein sequence comparison that yielded a hit in BLASTp, first, a scaled max score was calculated by dividing the largest max score by the sum of the lengths of the hit protein sequence and the reference protein sequence. For instance, the BLASTp query for *OXTR* yielded several hits in the tunicate *Ciona savignyi* and the hit with the highest max score was at a value of 92.8. This value was divided by 795, the sum of 364 (sequence length of said *OXTR* hit in *C. savignyi*) and 431 (sequence length of the reference *OXTR* protein in the sea lamprey), yielding a scaled max score of 0.1167296. The scaled max scores were then averaged across genes per target species, resulting in the scaled and averaged (mean) max score, which functioned as the cutoff value specific to a non-vertebrate species (i.e., each non-vertebrate species had its own unique cutoff value). If a scaled max score of a gene for a species was above that species-specific cutoff value, the hit was considered a putative homolog. As an example, the averaged, scaled cutoff value for *C. savignyi* is 0.2098398, hence the aforementioned *OXTR* BLASTp hit in *C. savignyi* with a scaled max score of 0.1167296 would not pass the threshold and would not be considered a homolog. These calculations allow one to normalize protein sequence lengths, which have an impact on sequence similarity discovery since BLASTp max scores for genes with longer sequences are by definition smaller than those for genes with shorter sequences due to BLASTp not accounting for sequence length (see also ref. 98). For all averaged, scaled max score thresholds and scaled max score values, see Supplementary Data 1, sheet 2. Orthologs and homologs were finally sorted into three categories. Genes dating back to species from phylostratum (PS) 1–3 ([99], including cellular organisms, eukaryota and opisthokonta) were classified as "ancient", genes with homologs dating back to PS 4–10 (including holozoa, metazoa, eumetazoa, bilateria, deuterostomia, chordata and olfactores) were classified as "medium-aged", and genes with orthologs in PS 11–20 (including among others vertebrata, gnathostomata, osteichthyes, tetrapoda, mammalia, and euarchontoglires) were classified as "modern" genes (see Supplementary Data 1, sheet 1, for a detailed phylogeny). We chose to primarily rely on max scores, since this metric allowed us to apply extended normalization statistics, unlike E-values, which we found not sufficiently informative on their own to make the homolog versus non-homolog differentiation. The phylogeny used to represent the evolutionary timeline from cellular organisms to humans including the species and lineage splits was constructed with reference to refs. 35–37,59,100–119 (but also note ref. 120) and verified with the help of the NCBI Taxonomy Browser (https://www.ncbi.nlm.nih.gov/Taxonomy/Browser/wwwtax.cgi).

### Signatures of natural selection in the jawless vertebrate and gnathostome OT pathway genes

A detailed guide with requirements, instructions, and tips for this positive selection analysis is provided in Supplementary Note 1. Briefly, exploratory tests for episodic positive selection across the same thirteen vertebrate species that were used in the BLASTp/microsynteny analyses (*H. sapiens*, *P. troglodytes*, *M. mulatta*, *M. musculus*, *B. taurus*, *D. novemcinctus*, *S. harrisii*, *O. anatinus*, *G. gallus*, *X. tropicalis*, *D. rerio*, *C. carcharias*, *P. marinus*), representing major branches in vertebrate evolution, were performed on the five genes that had their earliest ortholog in the extant proxy species (*P. marinus*) for the vertebrate ancestor (i.e., *ADCY9*, *GUCY1A2*, *OXTR*, *PIK3R5*, *PLA2G4A*) and on the 34 genes that had their earliest ortholog in the extant proxy species (*C. carcharias*) for the gnathostome ancestor (i.e., *ADCY1*, *ADCY2*, *ADCY7*, *ADCY8*, *CACNA2D1*, *CACNA2D3*, *CACN2D4*, *CACNG1*, *CACNG3*, *CACNG4*, *CACNG5*, *CD38*, *EEF2K*, *EGFR*, *ELK1*, *FOS*, *JUN*, *KCNJ12*, *KCNJ2*, *KCNJ3*, *KCNJ4*, *KCNJ5*, *MEF2C*, *MYLK3*, *NFATC1*, *NFACT2*, *NFACT3*, *OXT*, *PLA2G4F*, *PLCB1*, *PPP1R12A*, *PRKAB1*, *RAF1*, *RYR3*), as identified by the preceding combined BLASTp and microsynteny. First, orthologous protein sequences (AAS) and coding sequences (CDS) for the 39 genes in each of the 13 species—if an orthologous sequence was present—were manually downloaded from NCBI. The information on which sequences with which accession number were the

most credible was taken from the previous combined BLASTp and microsynteny analyses (compare Supplementary Data 2). For each gene, the AAS and CDS from the different species were manually collected into an aggregate protein FASTA file and a CDS FASTA file, respectively. This resulted in 39 protein FASTA and 39 CDS FASTA files. Subsequently, the AAS in each aggregative file were aligned using MUSCLE (version 5.1, https://github.com/rcedgar/muscle[121]) yielding 39 separate protein multiple sequence alignments, one for each gene. The protein multiple sequence alignments were translated into codon-based alignments based on the corresponding previously collected CDS using PAL2NAL (version 14[122]). The codon-alignment files were further processed for the main analysis by indicating missing orthologs with empty dummy sequences, by changing the sequence identifiers to shortened taxon names (e.g., "hsapiens", "ggallus"), and by converting the FASTA format codon-alignments to PHYLIP sequential format files using TriFusion (version 1.0.1, https://github.com/ODiogoSilva/TriFusion[123]). A vertebrate species tree was generated with TimeTree (version 5, http://timetree.org[124]), which draws on public data from published studies to generate reliable and robust species trees, and downloaded in Newick format. That tree is in line with the manually built tree derived from a literature search for the BLASTp and microsynteny analysis (compare to Fig. 1). The branch lengths in the species tree were manually removed from the tree file and the taxon names were shortened to match the sequence identifiers in the codon-based alignments. Following this, exploratory searches for positive selection of each gene along the vertebrate species tree using adaptive branch-site random effects likelihood models ("aBSREL", version 2.3[41]) from the HyPhy package (version 2.5.52, https://github.com/veg/hyphy[125]) were performed. aBSREL allows for non-hypothesis-driven (i.e., exploratory) positive selection analyses. That is, all branches and nodes are tested and no foreground branch needs to be specified a priori. The method uses a modified version of the classic branch-site model that allows for an adaptive number of $\omega$ classes per branch depending on the evolutionary properties of the branch. It compares the full, alternative model against a null model and then performs a Likelihood Ratio Test ("LRT", for a detailed description of the aBSREL test, see ref. 41). The method further comes with the advantage of automating running a large number of tests for multiple branches and nodes simultaneously. In R, all resulting raw $P$ values were corrected for multiple testing with the Benjamini–Hochberg False Discovery Rate (FDR) correction method at an initial significance threshold of $\alpha = 0.05$. The branch-site test did not finish for the gene *MYLK3*, most likely due to poor sequencing quality in the platypus *O. anatinus*.

## Functional annotation with FUMA

To functionally annotate the results from the phylostratigraphic BLASTp and microsynteny analyses, the data was submitted to the "Functional Mapping and Annotation of Genome-Wide Association Studies" (FUMA) web application (https://fuma.ctglab.nl,[45]) GENE2FUNC in three separate queries for the "ancient" (homologs in branches cellular organisms to opisthokonta, PS 1–3), "medium-aged" (homologs in branches holozoa to olfactores, PS 4–10) and "modern" (orthologs in branches vertebrata to homini, PS 11–20) gene sets (background set "Protein-coding", Ensembl version v92 (v102 for the ancient genes), MHC included, all default Bonferroni corrected, FDR correction for the gene-set enrichment testing). The tool uses RNA-seq data from GTEx v8[126] to test for tissue-specific enrichment of the gene set (here focused on 30 tissues across the human body, including the brain) and performs a hypergeometric test to assess enrichment of genes in different categories (i.e., GWAS) and pathways. Detailed information about the data sets and configuration of statistic tests implemented in FUMA can be found in ref. 45.

## Cerebral expression of modern genes supporting OT signaling

The FUMA tissue specificity results were extended with between-brain region expression specificity analyses for the modern genes supporting OT signaling. For these, first, expression data from the Allen Human Brain Atlas ("AHBA", $n = 6$ (1 female, 5 males), ages 24.0–57.0, three of Caucasian ethnicity, two African American, one Hispanic, https://human.brain-map.org[127]) was preprocessed with the Python toolbox abagen (version 0.1.1, https://github.com/rmarkello/abagen[47]) in Jupyter Lab based on the processing pipeline suggested by ref. 128. According to the documentation, the expression data collection process complied with the relevant ethical regulations concerning the collection and processing of human postmortem tissue samples. Consent from the next-of-kin of each donor was obtained. Detailed information regarding the data collection procedures can be found at https://human.brain-map.org and ref. 128 (see also Supplementary Information thereof). The following processing steps were performed for each of the six donors' cortical and subcortical regions in one joint query. Cortical and subcortical expression data for six brains was prepared using the 83-region (34 cortical, lateralized, and eight additional subcortical areas of which seven lateralized, brainstem structure bilateral) volumetric Desikan–Killiany atlas in MNI space[129] (subcortical structures included in the abagen analysis pipeline). Microarray probes were reannotated using data provided in ref. 128; probes not matched to a valid Entrez ID were discarded. Probes were then filtered based on their expression intensity relative to background noise[130]. Probes with intensity less than the background in less than 50.00% of samples across donors were excluded, yielding 31,569 probes. When multiple probes indexed the expression of the same gene, the probe with the most consistent pattern of regional variation across donors (i.e., differential stability[48]) was selected. Here, regions correspond to the structural designations provided in the ontology from the AHBA. The MNI coordinates of tissue samples were updated to those generated via non-linear registration using the Advanced Normalization Tools ("ANTs", https://github.com/chrisfilo/alleninf). Samples were assigned to brain regions in the Desikan–Killiany atlas if their MNI coordinates were within 2 mm of a given parcel. To reduce the potential of misassignment, sample-to-region matching was constrained by hemisphere and gross structural divisions (i.e., cortex, subcortex/brainstem, and cerebellum, such that e.g., a sample in the left cortex could only be assigned to an atlas parcel in the left cortex[128]). If a brain region was not assigned a tissue sample based on that procedure, every voxel in the region was mapped to the nearest tissue sample from the donor in order to generate a dense, interpolated expression map. The average of these expression values was taken across all voxels in the region, weighted by the distance between each voxel and the sample mapped to it, in order to obtain an estimate of the parcellated expression values for the missing region. Inter-subject variation was addressed by normalizing tissue sample expression values across genes using a robust sigmoid function[131]. Normalized expression values were then rescaled. Gene expression values were normalized across tissue samples using an identical procedure. All available tissue samples were used in the normalization process regardless of whether they were assigned to a brain region. Tissue samples not matched to a brain region were discarded after normalization. Samples assigned to the same brain region were averaged separately for each donor. These processing steps resulted in six expression matrices, one for each donor, with 83 rows corresponding to brain regions and 15,633 columns corresponding to genes (Supplementary Data 14–19, available at https://osf.io/rxphw). Finally, 81 modern genes in the OT pathway were extracted (17 were not available in the processed matrices). Differential stability (DS) values were also obtained with the abagen toolbox. Because the values are not directly accessible from the above-described workflow, they were calculated in an adjunct query with the difference that the DS estimates obtained with this procedure are computed as correlations between parcels in the Desikan–Killiany atlas as opposed to the main query where they are computed between the AHBA-defined brain structures. The DS values from the main query and the adjunct query can thus differ minimally, however, they generally are highly similar. Subsequently, between-brain region comparisons of the expression profile of the modern gene set in the OT pathway were calculated with the prepared matrices as follows. Between-region comparisons were implemented with 42 two-sided one-sample $t$ tests (one $t$ test per brain region, only left cortical hemisphere due to sample size limitations for the right hemisphere ($n = 2$)). For each brain region and each of the six subjects, the expression intensities of the

modern gene set facilitating OT signaling in that region were compared against the whole-brain population mean expression of the gene set (i.e., the average expression of the modern gene set facilitating OT signaling across all available brain regions and donors); *P* values were adjusted for 42 comparisons with the FDR correction method. The Desikan–Killiany cortical atlas values were visualized using the R package ggseg ([https://github.com/ggseg/ggseg](https://github.com/ggseg/ggseg)[132]). As a measure of effect size, Cohen's *d* for one-sample *t* tests was calculated.

## Reporting summary

Further information on research design is available in the Nature Portfolio Reporting Summary linked to this article.

## Data availability

Sequences used for the gene age estimations/phylostratigraphy and positive selection analysis were downloaded from [https://blast.ncbi.nlm.nih.gov](https://blast.ncbi.nlm.nih.gov). The databases for the functional annotation analysis in FUMA are included and available in the FUMA online tool at [https://fuma.ctglab.nl](https://fuma.ctglab.nl). The AHBA data and cerebral atlas used for this study are included in the Python toolbox abagen. Further information and other download links for the AHBA data are available at [https://human.brain-map.org](https://human.brain-map.org). All other data used in the analyses not mentioned above are deposited at [https://osf.io/rxphw](https://osf.io/rxphw)[133].

## Code availability

R and Jupyter Notebook/Python scripts used in the analyses with information on specific parameter settings, further code, data, Supplementary Information, and additional notes are all available at [https://osf.io/rxphw](https://osf.io/rxphw). A guide with requirements and instructions for the positive selection analysis is provided in Supplementary Note 1.

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

## Acknowledgements

This research was funded by the Research Council of Norway (301767) and the Novo Nordisk Foundation (NNF16OC0019856). Figure 1 and the phylogenetic tree in Fig. 3 were created in Adobe Illustrator 2023. Some of the icons in Fig. 1 were adapted from Biorender.com. Figures 2 and 4a were created with Biorender.com. The methodological report for the AHBA expression data preprocessing steps was generated with the `abagen` toolbox (https://github.com/rmarkello/abagen).

## Author contributions

A.M.S. and D.S.Q. conceived and planned the study. A.M.S. analyzed the data, with contributions from D.S.Q., C.T., J.R. and S.B. D.S.Q. and C.T. supervised the study. D.S.Q. provided funding for the study. A.M.S., D.S.Q., C.T., J.R., S.B., F.B., C.B., A.M.G.D.L., M.H., A.S., A.W., N.E.S., E.S., D.J.S., O.A.A., D.V.D.M. and L.T.W. contributed to the interpretation of the results. A.M.S. wrote the first and revised drafts of the manuscript, with D.S.Q., C.T., J.R., S.B., F.B., C.B., A.M.G.D.L., M.H., A.S., A.W., N.E.S., E.S., D.J.S., O.A.A., D.V.D.M. and L.T.W. contributing to the first and revised drafts of the manuscript.

## Competing interests

M.H. has served as a speaker for Lundbeck, outside of the work presented in the manuscript. The remaining authors declare no competing interests.

## Additional information

[1]Norwegian Centre for Mental Disorders Research (NORMENT), Institute of Clinical Medicine and Division of Mental Health and Addiction, University of Oslo and Oslo University Hospital, Oslo, Norway. [2]Department of Psychology, University of Oslo, Oslo, Norway. [3]Centre of Research and Education in Forensic Psychiatry, Oslo University Hospital, Oslo, Norway. [4]Faculty of Chemistry, Biotechnology and Food Science, Norwegian University of Life Sciences, Ås, Norway. [5]Natural History Museum, University of Oslo, Oslo, Norway. [6]Department of Medical Genetics, Division of Laboratory Medicine, Oslo University Hospital, Oslo, Norway. [7]Department of Psychiatric Research, Diakonhjemmet Hospital, Oslo, Norway. [8]Department of Clinical Neurosciences, Lausanne University Hospital (CHUV) and University of Lausanne, Lausanne, Switzerland. [9]Department of Psychiatry, University of Oxford, Oxford, UK. [10]Division of Mental Health and Addiction, Oslo University Hospital, Oslo, Norway. [11]Department of Mental Health and Suicide, Norwegian Institute of Public Health, Oslo, Norway. [12]Department of Child Health and Development, Norwegian Institute of Public Health, Oslo, Norway. [13]Hector Institute for Artificial Intelligence in Psychiatry, Central Institute of Mental Health, Medical Faculty Mannheim, Heidelberg University, Mannheim, Germany. [14]Department of Psychiatry and Psychotherapy, Central Institute of Mental Health, Medical Faculty Mannheim, Heidelberg University, Mannheim, Germany. [15]SA MRC Unit on Risk & Resilience in Mental Disorders, Department of Psychiatry and Neuroscience Institute, University of Cape Town, Cape Town, South Africa. [16]KG Jebsen Centre for Neurodevelopmental Disorders, University of Oslo and Oslo University Hospital, Oslo, Norway. [17]School of Mental Health and Neuroscience, Faculty of Health, Medicine and Life Sciences, Maastricht University, Maastricht, The Netherlands. [18]Rockefeller University, New York, NY, USA. [19]New York University, New York, NY, USA. [20]NevSom, Department of Rare Disorders, Oslo University Hospital, Oslo, Norway. ✉e-mail: ktheofanop@rockefeller.edu; daniel.quintana@psykologi.uio.no

