## [Peer Review File · Communications Biology]

Referee expertise:

Referee #1: Genetics, Evolution

Referee #2: oxytocin, social behaviour

Reviewers' comments:

Reviewer #1 (Remarks to the Author):

This manuscript describes the results of evolutionary analyses conducted on 154 genes that are involved in or support the oxytocin (OT) signaling pathway. The authors employ a variety of methods, including protein sequence similarity, microsynteny, and phylostratigraphy, to estimate the ages of these genes. They search for homologs or orthologs of OT pathway genes in the genomes of representative species from across 20 phylostrata, and estimate the emergence time of each gene, resulting in three main categories: ancient (28 genes), middle-aged (28 genes), and modern (98 genes). The authors then perform a gene set enrichment analysis using these three categories and identify intriguing significant enrichment patterns in various tissues, including muscle and brain. Following up on the gene set enrichment findings, they explore gene expression differences between brain regions in the "modern genes" subset.

In general, the rationale is well-described, and the background literature explains why OT pathway genes are important from an evolutionary perspective. The authors' use of a systematic approach to investigate such a complex gene pathway is a strength of this work. I have some questions/suggestions, that might improve the manuscript if clarified:

- The arrows and ordering of clades in Figure 1 could be misinterpreted as representing directional evolution towards the Homo clade. I recommend to replace Figure 1 with a simpler phylogenetic tree and to add branch split times, especially the times defining ancient, middle-aged, and modern genes.
- The majority of OT pathway genes seem to have their earliest orthologs in vertebrate species. It would be very interesting to see if these genes have been under selection in the vertebrate phylogeny (a software like PAML can be used for this: <http://abacus.gene.ucl.ac.uk/software/paml.html>).
- Figure 2 and Supplementary Materials 4a are almost identical. I would consider moving Figure 2 to supplementary to avoid repetition.
- It is intriguing that ~1/3 of modern OT pathway genes emerged in the gnathostomata/jawed vertebrate ancestor. However, as the authors also discuss, majority of the OT pathway genes are involved in multiple pathways and biological mechanisms. So I find it difficult to make any conclusions about the evolution of OT pathway or bone homeostasis only based on the microsynteny findings.
- It is noted that ancient and middle-aged gene sets each have 28 genes, while the modern gene set contains 98 genes. It is worth considering whether this difference in gene set sizes could be a concern for differences in enrichment analysis results.
- I find it difficult to follow the region-specific expression analysis of modern genes. Why almost every region is significant? Why the expression of OT pathway genes were compared against all protein coding genes instead of genes that are known to be expressed in the cortex or specific sub-cortical regions?

- All in all, the findings are rather descriptive. Paper can greatly improve if the authors clarify their hypotheses, test for selection on the OT pathway genes (at least for OXT, OXTR and CD38) in vertebrates (or another clade that they think is relevant), and refine their regional brain gene expression analysis.

Minor:

- line 96: sumateribstantiates → substantiates?
- Figure 5 is explained, but not cited in the Results section.

Reviewer #2 (Remarks to the Author):

The present manuscript created a comprehensive timeline of gene origins for all genes involved in oxytocin signaling and then related the 'age' of each gene to its expression pattern both within the brain and throughout the body. I lack the expertise to judge the statistics and methodological approach for this work, but I am equipped to comment on it as an oxytocin researcher. I found the work to be impressive in scope and presentation. The data visualization in particular was a strength of this paper. The figures were all very well put together.

However, I struggled with the breadth of the study's scope. In my view, too wide a net was cast when defining the oxytocin signaling pathway. The present study included 154 genes in this collection, but only a tiny fraction are specific to oxytocin signaling. Some, such as Actin or c-fos, are so fundamental to "basic, ubiquitous cellular mechanisms" that I fail to see a meaningfully specific connection to oxytocin. The stated goal of this paper is to "help inform the current purpose of the oxytocin pathway" but I don't see how the evolution of genes like Actin or c-fos could help inform my next oxytocin study. I am open to being convinced otherwise.

My more specific comments follow:

ABSTRACT

Line 27 "several processes" -while I appreciate the authors not pigeonholing oxytocin into a single function like social behavior, this current phrasing comes across a bit too broad. If word count space permits, it would be helpful to include an abbreviated list of oxytocin's most noteworthy functions.

Line 30 "purpose" implies a single role for oxytocin, which I don't think the authors intend; perhaps pluralize this word?

Line 31 "we assigned the genes" -which genes? Those of the OT signaling pathway?

INTRO

Line 43 - Again, the reader is left to their own as to why they should care about oxytocin since no detail is given to its roles in biology. A "diverse plethora" may be accurate but it is not descriptive nor compelling.

Line 53 - "parturition in annelids and possibly horses" is an odd framing. The role of oxytocin in parturition is well established for a number of mammalian species but this sentence makes it sound only tenuously investigated.

METHODS

This work identified 154 genes as belonging to the oxytocin signaling pathway. I wonder about the value of including so many fundamental subcellular signaling agents in this pathway. For instance, Gq alpha and the inositol trisphosphate receptor are included. I would not think of those as specific to the oxytocin signaling pathway since they are shared by so many other GPCRs. Would not each of those other receptor systems' evolution have acted on Gq alpha and IP3? And since so many other transmitters (dozens? a hundred perhaps?) all use these pathways, presumably their evolutionary histories would have contributed far more than that of oxytocin? This becomes even more the case when considering genes more distantly related to oxytocin. Included in the oxytocin signaling pathway are genes like Actin and c-fos, which I struggle to imagine a meaningful role for oxytocin in shaping their evolution. Actin is so fundamental a building block of all neurons it's often included as a housekeeping gene in expression studies. Among whatever forces contributed to the evolution of Actin, what fraction could oxytocin possibly account for? As an oxytocin researcher, am I expected to be guided by the evolution of Actin?

I would argue for a very limited scope when studying the evolution of the oxytocin system. Even genes such as CD38 seem only tangentially related to me. A few years ago I decided to educate myself on CD38 and was struck with just how predominantly oxytocin-unrelated its functions were. CD38 is an extremely pleiotropic molecule, with roles in immune functioning, cell adhesion, cancer, HIV, signal transduction and calcium signaling. This might be why for every CD38 paper that mentions oxytocin there are more than 200 that don't. Yes, CD38 regulates oxytocin, it also regulates countless other systems. We don't say "calcium regulates social behavior" because while it may be true, it's not specific enough to be a compelling research target.

RESULTS

Line 112 "OXT, OXTR, CD38 and the secondary genes" – I am confused as to the definition of "secondary genes" here. Earlier, "secondary" was used in reference to genes relating to fundamental processes such as calcium voltage-gated channels outside of the 154 genes of the oxytocin pathway. The phrasing here implies that secondary means something different.

Figure 4 – How should readers interpret the presence of OXT mRNA in these subcortical brain regions given that oxytocin is not synthesized in these regions?

Authors: We thank the reviewers for their helpful feedback. We believe that the manuscript has been noticeably improved after implementing the reviewer's feedback. The tracked changed PDF version of the manuscript is best viewed in Adobe Acrobat.

Reviewer #1 (Remarks to the Author):

This manuscript describes the results of evolutionary analyses conducted on 154 genes that are involved in or support the oxytocin (OT) signaling pathway. The authors employ a variety of methods, including protein sequence similarity, microsynteny, and phylostratigraphy, to estimate the ages of these genes. They search for homologs or orthologs of OT pathway genes in the genomes of representative species from across 20 phylostrata, and estimate the emergence time of each gene, resulting in three main categories: ancient (28 genes), middle-aged (28 genes), and modern (98 genes). The authors then perform a gene set enrichment analysis using these three categories and identify intriguing significant enrichment patterns in various tissues, including muscle and brain. Following up on the gene set enrichment findings, they explore gene expression differences between brain regions in the "modern genes" subset.

In general, the rationale is well-described, and the background literature explains why OT pathway genes are important from an evolutionary perspective. The authors' use of a systematic approach to investigate such a complex gene pathway is a strength of this work. I have some questions/suggestions, that might improve the manuscript if clarified:

Author response: We thank the reviewer for the summary of the manuscript.

- The arrows and ordering of clades in Figure 1 could be misinterpreted as representing directional evolution towards the Homo clade. I recommend to replace Figure 1 with a simpler phylogenetic tree and to add branch split times, especially the times defining ancient, middle-aged, and modern genes.

Author response 1.1: We adjusted the figure according to the reviewer's suggestions. That is, we created a simplified version, removed the arrows, added branch split times and highlighted the three different gene age categories by different shades of grey (shown below but also see page 3, figure 1).

- The majority of OT pathway genes seem to have their earliest orthologs in vertebrate species. It would be very interesting to see if these genes have been under selection in the vertebrate phylogeny (a software like PAML can be used for this: <http://abacus.gene.ucl.ac.uk/software/paml.html>).

Author response 1.2: Thank you for this important suggestion. Following this suggestion, we implemented an exploratory positive selection analysis to investigate whether the genes with their earliest ortholog in the vertebrate ancestor and gnathostome ancestor in the OT pathway have been under specific selective pressures during vertebrate evolution. To do this, we downloaded relevant protein and nucleotide sequences from NCBI using the information on orthology from our BLASTp/microsynteny analysis, we used MUSCLE to align them and PAL2NAL to translate them to codon-based alignments. We generated a vertebrate species tree with TimeTree (<http://timetree.org/>) as the trees generated with ORTHOFINDER were unreliable and not accurate (e.g., they suggested that the amniota emerged before tetrapoda, perhaps due to the wide range of diverged branches and years spanned). We then used aBSREL (“adaptive branch-site random effects likelihood”) from HyPhy (Smith et al., 2015; <https://doi.org/10.1093/molbev/msv022>) to test for positive selection of the relevant genes. We chose aBSREL from HyPhy over PAML because aBSREL allows for the simultaneous *exploratory* testing of all branches/nodes without the need to specify of a foreground branch. The description of the methods is in the revised manuscript (page 13, line 517-565), and a detailed manual describing the steps to replicate the analysis with tips and comments for the naïve user is provided in supplementary material 13. The analysis results are described (page 4, line 157-201) and discussed (page 9, line 305-341; page 11, line 397-403) in the updated manuscript as well. We added figure 3 (shown below,

but also see page 6 in the revised manuscript) to visualize the max. dN/dS value distribution from all aBSREL tests.

- Figure 2 and Supplementary Materials 4a are almost identical. I would consider moving Figure 2 to supplementary to avoid repetition.

Author response 1.3: We agree with the reviewer that figure 2 and the figure in the supplementary materials (SM) 4a are similar, and at the same time we believe that the manuscript benefits from an in-text visual representation of the microsynteny analyses for one exemplary gene (here, *OXT*). We therefore implemented the following alternative solution, considering the reviewers remark on redundancy: We removed the original SM 4a, kept figure 2 in the text as is, and replaced SM 4b (now 4a) and SM 4c (now 4b) by microsynteny analyses with the same reduced level of detail as in-text figure 2 (page 5, figure 2; supplementary materials 4 a and b).

- It is intriguing that ~1/3 of modern OT pathway genes emerged in the gnathostomata/jawed vertebrate ancestor. However, as the authors also discuss, majority of the OT pathway genes are involved in multiple pathways and biological mechanisms. So I find it difficult to make any conclusions about the evolution of OT pathway or bone homeostasis only based on the microsynteny findings.

Author response 1.4: We now put more emphasis on these conclusions that further research would be needed to make more definite statements about these conclusions (page 8, line 287-290):

“Whether these OT functions markedly shaped gnathostome evolution or whether the observation is coincidental needs to be further substantiated by more research, as microsynteny analyses alone cannot determine this..”

- It is noted that ancient and middle-aged gene sets each have 28 genes, while the modern gene set contains 98 genes. It is worth considering whether this difference in gene set sizes could be a concern for differences in enrichment analysis results.

Author response 1.5: The reviewer raises an important point here. We double-checked the methods behind FUMA GENE2FUNC and agree that with hyper-geometric tests, smaller gene sets are more likely to have false positives. Since FUMA GENE2FUNC is a widely applied tool, we checked other papers that use the tool [e.g., Vogelezang et al., 2022 (<https://doi.org/10.1186/s12920-022-01281-1>); Alonso-Gonzalez et al., 2019 (<https://doi.org/10.1186/s12920-019-0593-5>); Zhang et al., 2022 (<https://doi.org/10.1097/ADM.0000000000001112>); Sreiretnakumar et al., 2019 (<https://doi.org/10.1016/j.schres.2019.04.026>)] and what they did to evaluate this potential issue. None of the studies, however, addressed it in their work, not to mention performed additional analyses, and there don't seem to be correction methods commonly applied. We therefore, considering the current status quo in the literature, decided to include the following sentence in the limitations to address this potential issue (page 11, line 403-410): *„Concerning the FUMA analysis, we submitted three gene sets of varying sizes (n = 28, n = 28, n = 98). Since the online tool is based on hyper-geometric tests operating in the background, it must be noted that smaller gene sets, like the ancient and medium-aged gene sets in this study, are more likely to produce false positives, as compared to larger gene sets, such as the modern gene set in this study. Looking at the accumulated published studies using FUMA, there currently does not seem to be a standard corrective mechanism for this potential issue in place. The investigation of such a mechanism may be of interest to future studies.“*

- I find it difficult to follow the region-specific expression analysis of modern genes. Why almost every region is significant? Why the expression of OT pathway genes were compared against all protein coding genes instead of genes that are known to be expressed in the cortex or specific sub-cortical regions?

Author response 1.6: We assume that the reviewer is referring to the analysis visualized in former, original figure 5 in the manuscript. We agree with the reviewer that it is notable that almost every region is significant, however after double-checking the data and the analysis, the result was same. Thus, to further investigate the nature of the result, we ran additional analyses with the ancient (R script “rev_fig05_ancient_genes.R”) and the medium-aged (R script “rev_fig05_mediumaged_genes.R”) gene sets to check whether the observed expression pattern is something that is unique to the modern genes or whether it is a generic form of cerebral gene expression [All R scripts, figures and tables mentioned in this paragraph are available on OSF (<https://osf.io/rxphw/>), in the folder “Revisions and archive > “revision_analysis_original_fig05”]. As can be seen in the additional figures and tables “rev_mag_fig05”/“rev_mediumaged_genes.xlsx” for the medium-aged and “rev_anc_fig05”/“rev_ancient_genes.xlsx” for the ancient genes, the expression pattern for the other two gene subsets is very similar to the expression pattern of the modern gene set. Thus, we decided to remove the analysis underlying the original figure 5 from the manuscript since it seemed to not be robust.

- All in all, the findings are rather descriptive. Paper can greatly improve if the authors clarify their hypotheses, test for selection on the OT pathway genes (at least for OXT, OXTR and CD38) in vertebrates (or another clade that they think is relevant), and refine their regional brain gene expression analysis.

Author response 1.7: Since the study was exploratory in nature, we could not declare any hypotheses. We now clarify in the introduction and discussion that the study was exploratory, which is why no *a priori* hypotheses were stated (page 3, line 95-96; and page 7, line 250-251):

“In this exploratory study, we applied a phylostratigraphic approach, using protein [...]”
and

“With our open, exploratory approach and analyses, we identified the evolutionary time points when [...]”.

However, we now discuss the results in the discussion in light of the existing hypotheses from previous research and how the results fit in the larger literature based on the knowledge in the current field. Furthermore, we added the test for positive selection for the 39 candidate genes (see response 1.2 above for details) and reframed the cerebral expression analyses as described above (response 1.6) so that it is more robust and sounder.

Minor:

- line 96: sumateribstantiates → substantiates?

Author response 1.8: We corrected that typo. Thank you for pointing it out.

- Figure 5 is explained, but not cited in the Results section.

Author response 1.9: See response 1.6 above for details. The analysis and former, original corresponding figure 5 are removed.

.....
Reviewer #2 (Remarks to the Author):

The present manuscript created a comprehensive timeline of gene origins for all genes involved in oxytocin signaling and then related the ‘age’ of each gene to its expression pattern both within the brain and throughout the body. I lack the expertise to judge the statistics and methodological approach for this work, but I am equipped to comment on it as an oxytocin researcher. I found the work to be impressive in scope and presentation. The data visualization in particular was a strength of this paper. The figures were all very well put together.

Author response: We thank the reviewer for the summary of the manuscript.

However, I struggled with the breadth of the study's scope. In my view, too wide a net was cast when defining the oxytocin signaling pathway. The present study included 154 genes in this collection, but only a tiny fraction are specific to oxytocin signaling. Some, such as Actin or c-fos, are so fundamental to "basic, ubiquitous cellular mechanisms" that I fail to see a meaningfully specific connection to oxytocin. The stated goal of this paper is to "help inform the current purpose of the oxytocin pathway" but I don't see how the evolution of genes like Actin or c-fos could help inform my next oxytocin study. I am open to being convinced otherwise.

My more specific comments follow:

ABSTRACT

Line 27 "several processes" -while I appreciate the authors not pigeonholing oxytocin into a single function like social behavior, this current phrasing comes across a bit too broad. If word count space permits, it would be helpful to include an abbreviated list of oxytocin's most noteworthy functions.

Author response 2.1: While the word count for the abstract is very strict, we adjusted the phrasing to include two examples (page 1, line 28-29):

"Oxytocin (OT) is a neuropeptide associated with both psychological and somatic processes like parturition and social bonding."

Line 30 "purpose" implies a single role for oxytocin, which I don't think the authors intend; perhaps pluralize this word?

Author response 2.2: Due to word count constraints and adding new information to the abstract based on the new analysis that reviewer #1 requested, the sentence in question has been cut out.

Line 31 "we assigned the genes" - which genes? Those of the OT signaling pathway?

Author response 2.3: We added additional information for clarification (page 1, line 31-33):

"Using protein sequence similarity searches, microsynteny, and phylostratigraphy, we assigned the genes supporting the OT pathway to different phylostrata based on when we found they likely arose in evolution."

INTRO

Line 43 – Again, the reader is left to their own as to why they should care about oxytocin since no detail is given to its roles in biology. A "diverse plethora" may be accurate but it is not descriptive nor compelling.

Author response 2.4: A few examples of the functions of the oxytocin system with referral to the more detailed discussion below in the paragraph is now given (page 2, line 44-46):
“Oxytocin (OT) is a hormone and neuromodulator involved in a diverse plethora of functions in the central and peripheral nervous system across a wide range of species, such as parturition, metabolism, or social cognition (as detailed below).”

Line 53 – “parturition in annelids and possibly horses” is an odd framing. The role of oxytocin in parturition is well established for a number of mammalian species but this sentence makes it sound only tenuously investigated.

Author response 2.5: We rephrased the sentence so that the well-established role of OT in parturition across many different species reads clearer and more robust (page 2, line 55-59):
“For instance, the involvement of OT in parturition has been demonstrated not only in a vast number of mammalian species (e.g., non-human primates (Vargas-Pinilla et al., 2015), rabbits (Fuchs & Dawood, 1980), and many other (Landgraf et al., 1983, including vasotocin; Rapacz-Leonard et al., 2020; Gram et al., 2014)), but possibly also in non-vertebrates such as annelids in the form of what has been considered distant homologs (Fujino et al., 1999).”

METHODS

This work identified 154 genes as belonging to the oxytocin signaling pathway. I wonder about the value of including so many fundamental subcellular signaling agents in this pathway. For instance, Gq alpha and the inositol trisphosphate receptor are included. I would not think of those as specific to the oxytocin signaling pathway since they are shared by so many other GPCRs. Would not each of those other receptor systems' evolution have acted on Gq alpha and IP3? And since so many other transmitters (dozens? A hundred perhaps?) all use these pathways, presumably their evolutionary histories would have contributed far more than that of oxytocin? This becomes even more the case when considering genes more distantly related to oxytocin. Included in the oxytocin signaling pathway are genes like Actin and c-fos, which I struggle to imagine a meaningful role for oxytocin in shaping their evolution. Actin is so fundamental a building block of all neurons it's often included as a housekeeping gene in expression studies. Among whatever forces contributed to the evolution of Actin, what fraction could oxytocin possibly account for? As an oxytocin researcher, am I expected to be guided by the evolution of Actin?

I would argue for a very limited scope when studying the evolution of the oxytocin system. Even genes such as CD38 seem only tangentially related to me. A few years ago I decided to educate myself on CD38 and was struck with just how predominantly oxytocin-unrelated its functions were. CD38 is an extremely pleiotropic molecule, with roles in immune functioning, cell adhesion, cancer, HIV, signal transduction and calcium signaling. This might be why for every CD38 paper that mentions oxytocin there are more than 200 that don't. Yes, CD38 regulates oxytocin, it also regulates countless other systems. We don't say "calcium regulates social behavior" because while it may be true, it's not specific enough to be a compelling research target.

Author response 2.6: The reviewer raises some interesting points with their comment. To begin with we would like to reiterate that we did not identify or pick the 154 genes as part of the OT system. We retrieved that information from a recognized consensus encyclopedia, namely 'KEGG' (Kyoto Encyclopedia of Genes and Genomes), which attributes the 154 genes to the pathway. Since this may possibly not have been evident from the manuscript, we added additional information clarifying how the genes were selected (page 2, line 74-78; page 12, line 439-442):

"However, more than 150 other genes are associated with the OT pathway as they enable and support OT signaling and functions, and mediate OT's effects on further agents and pathways like MAP kinases in humans, as reported in the established consensus encyclopedia on genes, genomes and pathways, "KEGG" (Kanehisa et al., 2017; Devost et al., 2008; detailed official pathway information available at <https://www.genome.jp/entry/pathway+hsa04921>)."

And

"The 154 genes thought to be involved in OT signaling were retrieved from the official consensus encyclopedia "Kyoto Encyclopedia of Genes and Genomes" (KEGG; Kanehisa et al., 2017), where they are reported to be part of the OT signaling pathway. An overview of the pathway information and visual representation can be viewed at <https://www.genome.jp/entry/pathway+hsa04921>, see also Devost et al., 2008."

We would also like to add that the entry in KEGG is based on the published work by Devost et al. (2008; [https://doi.org/10.1016/S0079-6123\(08\)00415-9](https://doi.org/10.1016/S0079-6123(08)00415-9)), which is also cited in the manuscript. Briefly, Devost and colleagues explore the different signaling pathways of the OT system. As the Devost and colleagues underscore, the OT receptor gene is expressed in a variety of tissues and is marked by diverse functions. This diversity seems to be reflected in the pathways (facilitated by genes) that the OT system includes, uses and activates. For instance, the OT receptor seems to work in concert with different MAP kinases in the myometrium. This widened scope is relevant for understanding OT's role in parturition, for instance. In their work the authors also reveal new pathways of the hormone system involving for instance *ERK5* and *EEF2*. We believe that this polygenic, holistic approach may help shed more light on the wide-ranging functions of the OT system, which go well beyond social behavior. Rather than focusing on the isolated actions of three candidate genes, identifying further pathways and systems the genes activate and govern may be more informative. We can understand the reviewer's feedback on the evolution of for instance Actin and in how far that may be relevant to oxytocin research. As mentioned above, we used a consensus encyclopedia to identify our genes of interest, which is a more objective approach than picking and choosing putatively relevant OT system genes ourselves. Selectively removing genes like *ACTG1* or *ACTB* from the oxytocin pathway may introduce bias to our evidence-based approach.

RESULTS

Line 112 "OXT, OXTR, CD38 and the secondary genes" – I am confused as to the definition of "secondary genes" here. Earlier, "secondary" was used in reference to genes relating to fundamental processes such as calcium voltage-gated channels outside of the 154 genes of

the oxytocin pathway. The phrasing here implies that secondary means something different.

Author response 2.7: We agree that the use of the word “secondary” in the different contexts might be confusing. Therefore, we rephrased the sentence in different locations in the text and hope they now imply the same meaning (page 2, line 78-79; page 4, line 120; page 7, line 274-275):

“Among the sets of associated genes in the OT pathway is [...]”

and

“PSS searches among genes facilitating OT signaling revealed [...]”

and

“Interestingly, subunits CACNG2-4, 7 and 8, along with OXT and OXTR and other genes in the OT pathway [...]”

Figure 4 – How should readers interpret the presence of OXT mRNA in these subcortical brain regions given that oxytocin is not synthesized in these regions?

Author response 2.8: We double-checked our raw expression data from the Allen Human Brain Atlas that underlies part A of figure 5 (originally 4, now figure 5) and can confirm that in our raw data, *OXT* mRNA can be found in the displayed brain areas. Of note, among the group of ‘modern’ genes, the *OXT* mRNA signal is only in the mid-range of intensity in the majority of sub-cortical distributions. That means in the putamen, hippocampus, amygdala, pallidum, and brainstem the *OXT* mRNA signal is around the mean of the distributions, in the caudate it is below the mean, and only in the thalamus proper and accumbens area is the *OXT* mRNA signal around medium-to-high (cf. supplementary materials 11b). We also browsed the online visualization tool from the Allen Institute in case something with our version of the raw data was incorrect and found mRNA signal for *OXT* all over the brain to varying degrees here, too (<https://human.brain-map.org/microarray/search/>). We further compared the gene expression data from the Allen Human Brain Atlas with other gene expression databases (i.e., Bgee (<https://www.bgee.org/>), GTEX (<https://gtexportal.org/home/>)). For instance, in the “Gene expression data in animals” (‘Bgee’) database, the entry for *OXT* in the modern human shows mRNA signal in the pallidum, the caudate, the thalamus, and the hippocampus, amongst other regions outside the hypothalamus (all available at <https://www.bgee.org/>). Furthermore, other peer-reviewed published studies also find mRNA signal for *OXT* in different brain regions other than the hypothalamus (Quintana et al., 2021, see the supplementary information of this article). The sub-cortical atlas that came with the Desikan-Killiany atlas in combination with the abagen toolbox which we used for parcellating the expression data, does not include the hypothalamus, so we cannot directly compare the expression intensities of all different subcortical regions in our parcellated data. However, the mRNA signal intensity of *OXT* in regions where it is expected to be highly expressed (e.g., hypothalamus) is, according to the Allen Human Brain Atlas web browser, compared to the other sub-cortical regions, still substantially higher.

Thus, the presence of some mRNA in other brain regions might be expected, and despite that presence in other sub-cortical regions, the overall region-specific intensities still are in line with the literature. The observation that small amounts of *OXT* mRNA can be found in those brain regions hence does not seem to be a result of our analyses but rather an interesting naturally occurring phenomenon across different datasets. Still, we can speculate as to why this happens.

It could be that the precursor protein that the *OXT* gene encodes is transcribed region-unspecific, and that the synthesis of oxytocin from the precursor is then region-specific. Furthermore, the observed phenomenon is not unique to *OXT*. For instance, some amounts of *DRD2* mRNA can be found in brain regions where one would not necessarily “expect” it (e.g., hippocampus, inferior temporal gyrus, frontal lobe, cf. <https://human.brain-map.org/>) given its main synthesis sites (i.e., striatum, substantia nigra, ventral tegmental area (Khlghatyan et al., 2019, <https://doi.org/10.1093/cercor/bhy261>)). Similar patterns can be found for the gene *SLC6A4*, of which the transporter is predominantly synthesized in the midbrain tegmentum (esp. raphe nucleus, ventral tegmental area), pons, different subunits of the medulla, however, traces of mRNA can also be found in other brain regions (e.g., amygdala, frontal lobe, hippocampal formation, cf. <https://human.brain-map.org/>).

We added a note to the discussion of the results from the cerebral expression analysis, pointing out the above-elaborated circumstance (page 10, line 364-369):

“We also found some OXT mRNA signal in subcortical regions that are usually not prominently associated with the synthesis of OXT. However, the presence of small amounts of OXT mRNA in those regions is also reported in other databases (e.g., “Bgee”, Bastian et al., 2021, <https://www.bgee.org/>; “GTEx”, Lonsdale et al., 2013, <https://gtexportal.org/>) and studies (e.g., Quintana et al., 2019, using the same data), and the OXT mRNA intensity is still considerably higher in regions where it is to be expected (e.g., hypothalamus, compare <https://human.brain-map.org/>).”

REVIEWERS' COMMENTS:

Reviewer #1 (Remarks to the Author):

I would like to thank the authors for thoroughly addressing each of the raised points. The manuscript greatly improved with the addition of a positive selection analysis, the exclusion of brain region-specific gene expression analysis results and former Fig. 5, clarification of the exploratory nature of this work, and discussion of limitations in more detail. It is intriguing to see in the new results that the dN/dS distribution gets narrower as the branch age gets younger, which might worth looking into in another study.

While the previous version of Fig. 1 was more visually appealing, I find this revised Fig. 1 to be clearer and more accessible to a general audience, marking another improvement for the paper. I also appreciate that the authors ensured the accessibility of their raw data, scripts and results, which facilitates reproducibility. I have no additional comments and recommend this manuscript for publication.

Reviewer #2 (Remarks to the Author):

I sincerely appreciate the authors' responses to my questions and concerns. From my perspective, the manuscript has been improved as a result.